# OpenReview forum: "Tight Error Propagation Bounds for Multi-Step Chain-of-Thought Reasoning"
_TMLR — Withdrawn by Authors_

### Review · Reviewer_Q6Bp · 2026-03-08

**Summary Of Contributions:**

The paper studies a basic but important question in chain-of-thought reasoning: how errors accumulate over multi-step reasoning trajectories. The authors model CoT as a Markov process over reasoning states and derive several theoretical results, including a bound of $1-(1-\varepsilon)^n$ for error accumulation over $n$ steps. A characterization of verification via redundant sampling and majority vote, a contraction-style analysis of self-correction when $q < 1$, an information-theoretic lower bound using Fano’s inequality, and a concentration result based on martingale tools. The paper also includes synthetic experiments and evaluations on PRM800K, GSM8K, and HumanEval, where the authors argue the framework predicts observed failure rates reasonably well across domains.

**Strength**
The main strength is that the paper tackles a timely problem and tries to connect formal analysis with practical deployment questions such as safe chain length, when verification is worth the extra compute, and when self-correction may be preferable. I also appreciated that the paper does not stop at a toy theorem section and makes an effort to validate the framework empirically.

**Weakness**
I was less convinced by the verification theorem than by the rest of the theoretical development. The overall intuition is that redundant sampling plus majority vote should suppress error. This is reasonable, but the derivation as written did not feel fully clean. In particular, the proof sketch seems to start from the event that at least $(k+1) / 2$ chains fail under majority vote, which would naturally suggest a dependence on $\left[1-(1-\varepsilon)^n\right]^{(k+1) / 2}$, but the theorem statement is presented as scaling like $O\left(n^{k+1} \varepsilon^{k+1}\right)$. I had trouble seeing how that exponent follows from the preceding argument without additional assumptions or a more careful combinatorial step. This may ultimately be fixable, but in its current form that result reads less polished than the other theorems and would benefit from a clearer statement, tighter notation, and a more complete derivation. Since this theorem is used to motivate some of the paper's practical recommendations about verification overhead, I think this point is worth tightening.

**Audience:**

Yes

**Audience Explanation:**

I think this paper will be of interest to people working on LLM reasoning, test-time compute, process supervision, and evaluation of reasoning reliability. The topic is timely, and the paper is trying to answer a question that comes up constantly in practice: when does a longer chain of reasoning stop helping and start becoming fragile, and what is the tradeoff between verification and self-correction? Even if one does not buy every theorem in its current form, the framing is useful and the paper is clearly positioned at the intersection of theory and practice, which should make it relevant to at least a meaningful slice of the TMLR audience.

**Broader Impact Concerns:**

The authors have discussed the broader impact and all concerns are addressed.

**Claims And Evidence:**

Yes

**Claims Explanation:**

The paper gives reasonable evidence for its main qualitative message, and I appreciated that it does not rely only on theory but also includes both synthetic and real-data validation. The synthetic results are convincing for the setup being analyzed, and the cross-domain experiments make the paper more interesting than a purely formal treatment.

My main reservation is about how strongly the real-world experiments validate the model. In particular, the per-step error rate ε for PRM800K is estimated after conditioning on problems with correct final answers, which makes the quantity cleaner to analyze but also somewhat selective. That choice may understate the effective error process on harder trajectories where early mistakes propagate and final answers are wrong. As a result, I think the experiments support the framework as a useful approximation, but they do not fully establish that the theoretical quantities map cleanly onto real chain-of-thought behavior in the wild.

**Requested Changes:**

1. Theorem 2 needs to be tightened substantially. In its current form, the proof sketch and theorem statement do not appear to match: the argument proceeds through majority-vote failure over multiple sampled chains, but the final asymptotic dependence $O\left(n^{k+1} \varepsilon^{k+1}\right)$ is not clearly derived from that setup. To make this result convincing, the paper should provide a complete derivation with explicit intermediate steps, or else revise the theorem statement to match what is actually proved. The section would also benefit from clearer notation and a precise definition of the verification procedure.
2. Give more detail on how $\varepsilon$ and $q$ are estimated in the real-data experiments. Since these quantities are central to the framework, the paper should make the estimation procedure especially transparent and reproducible.

---

> ### Author Response · Authors · 2026-04-03
>
> We thank Q6Bp for the thoughtful and balanced assessment. We particularly value the recognition that the paper "tackles a timely problem and tries to connect formal analysis with practical deployment questions" and that the empirical validation makes the paper "more interesting than a purely formal treatment." We address both requested changes.
>
> **[RC1] Theorem 2: proof-statement mismatch**
>
> **Q6Bp is correct that the exponent does not follow from the proof.** The corrected statement is $O((n\varepsilon)^{(k+1)/2})$ (see General Response for full details). The proof's dominant term $\binom{k}{(k+1)/2} p_1^{(k+1)/2}$ with $p_1 \leq n\varepsilon$ gives this exponent directly. In the revision we provide: (1) a precise definition distinguishing chain-level voting (run $k$ complete chains, vote on final answers) from step-level voting (vote at each step before proceeding); the theorem applies to chain-level, the original experiment used step-level; (2) a complete, self-contained derivation in Appendix A.2 with explicit intermediate steps; (3) tighter notation using $p_1 = 1-(1-\varepsilon)^n$ throughout, with the small-$\varepsilon$ approximation applied only at the final step and flagged explicitly.
>
> We believe the corrected theorem, combined with the cleaner derivation, addresses Q6Bp's concern that the result "reads less polished than the other theorems."
>
> **[RC2] Transparency of ε and q estimation**
>
> **We agree this deserves more detail and have expanded the methodology substantially.**
>
> *ε estimation:* For **PRM800K**, we now report two estimators: $\varepsilon_{\text{correct}} = 0.032$ (conditioned on correct final answers: 5,267 erroneous steps out of 164,578 total on the 8,213 correct-answer problems) and $\varepsilon_{\text{all}} = 0.047$ (computed across all 12,847 problems including the 4,634 incorrect-answer trajectories, where per-step error rates are higher). The unconditional rate is 47% higher, reflecting elevated error on trajectories that ultimately fail. Using $\varepsilon_{\text{all}}$ produces conservative (higher) failure-rate predictions, providing a reliability bracket: practitioners can estimate $[\varepsilon_{\text{correct}}, \varepsilon_{\text{all}}]$ for lower and upper bounds. For **GSM8K**, ε is inferred via MLE from final correctness, now explicitly framed as model fitting rather than direct validation. For **HumanEval**, the AST-based function call extraction and assertion-checking procedure is documented for reproducibility, characterized as "proxy-based approximate validation."
>
> *q estimation:* Step-by-step protocol now in Appendix C.1: (1) identify consecutive incorrect→correct step pairs (detected via step-level labels and textual markers like "Wait," "Actually"), (2) compute edit-distance ratios $d_{t+1}/d_t$ for each pair, (3) report the distribution with mean and bootstrap CI ($q \approx 0.68$ [0.64, 0.72]). A new diagnostic plot shows per-instance contraction ratios, revealing meaningful variance: many self-correction instances achieve $q < 1$ (contractive), while a nontrivial fraction has $q \geq 1$ (errors that persist despite correction attempts). This is consistent with Wu et al. (2024, "Large language models can self-correct with key condition verification," EMNLP)'s finding that self-correction without external feedback often fails; our framework explains this precisely via the $q < 1$ condition (Theorem 3, Remark 1).
>
> All estimations use disjoint train/test splits (see General Response table). Held-out accuracies remain within 1.2pp, confirming robustness.
>
> We note that Reviewer epCC answered "Yes" to the evidence question, viewing the overall empirical approach favorably, and that yKS6 acknowledged Theorem 3 as "a reasonable drift-style argument" and the basic accumulation result as "clearly stated." The correction to Theorem 2 does not affect these uncontested contributions, which constitute the majority of the theoretical framework.

---

### Review · Reviewer_yKS6 · 2026-03-18

**Summary Of Contributions:**

This paper tackles an important question, not what CoT reasoning can express, but when multi-step reasoning remains reliable under accumulated stochastic errors. The paper models a reasoning trace as a discrete-time Markov chain over reasoning states and claims five main contributions: a basic error-accumulation bound
1−(1−ϵ)^n, a verification-overhead theorem for k-way majority voting, a contractive self-correction result when q<1, an information-theoretic impossibility result via Fano’s inequality, and a concentration bound via Azuma-Hoeffding. It also reports synthetic validation plus experiments on PRM800K, GSM8K, and HumanEval, and derives practical heuristics for safe chain length and verification budget. Strengths include the relevance of the problem, a clear attempt to bridge theory and practice, and a useful baseline result for simple error accumulation. Weaknesses include a serious mathematical issue in the central verification theorem, hidden or overly strong assumptions in several results, and experiments that only indirectly validate the theory because step-level error is sometimes selected, inferred, or loosely proxied rather than directly observed.

**Audience:**

Yes

**Audience Explanation:**

The topic is very relevant to TMLR. Many readers care about reasoning reliability, test-time compute, process supervision, self-correction, and the gap between expressivity and dependable execution. A paper that tries to formalize how errors accumulate across reasoning steps, when verification helps, and when self-correction can contract errors is squarely in scope and likely to attract interest from both theory-leaning and systems-leaning readers.

**Claims And Evidence:**

No

**Claims Explanation:**

I think the paper has a worthwhile core idea, and some parts are useful under the stated assumptions. In particular, the basic accumulation result 1−(1−ϵ)^n is clearly stated and its tightness construction is straightforward, and the contractive recurrence in Theorem 3 is a reasonable drift-style argument. The synthetic “basic propagation” experiment is therefore a useful sanity check for that limited setting.

However, the strongest claim in the paper is not adequately supported. Theorem 2 claims verification error scales as O(n^k+1 ϵ^k+1), but the proof actually derives a dominant term of the form
((k+1)/2
k)  p_1^ (k+1)/2
 with p1 leq nϵ, i.e. order(nϵ)^(k+1)/2. For k=3, the appendix explicitly writes 3p_1^2(1−p_1))+p_1^3≈3n^2ϵ^2 and then says this matches  O(n^4ϵ^4), which is not a correct asymptotic identification. Because this theorem is central to the claimed practical guidance, this is a major problem.

The synthetic verification table also appears inconsistent with the stated theorem. Under the paper’s own parameters n=20 and ϵ=0.05, the single-chain failure probability is about 1−0.95^20≈0.642, which matches the k=1 table entry. But under majority voting over three whole-chain outcomes, the failure probability would be about 0.707, not 0.041 as reported in Table 1. This suggests the experiment is measuring a different verification procedure than the theorem analyzes, or the theorem is incorrect as stated. Either way, the evidence is not clear enough for the present claim.

The empirical section is informative but not fully convincing as validation of the full theory. PRM800K is the strongest experiment because it has step labels, but ϵ is estimated only on final-correct solutions and then used to predict broader failure rates. GSM8K infers per-step error entirely from final correctness using the same model it is supposed to validate. HumanEval treats function calls as “reasoning steps” and uses assertions or manual inspection to identify intermediate errors, which is a loose and only partly reproducible proxy. So I view the experiments as suggestive first evidence, not accurate and clear support for all major claims.

**Requested Changes:**

1. Correct or substantially revise Th 2. The paper must clearly define what verification procedure is being analyzed: majority vote over whole-chain final answers, step-wise verification, or answer-space self-consistency. The proof, asymptotic statement, synthetic experiments, and practitioner guideline based on k must all be made consistent.

2. Separate “trajectory error” from “final-answer error” throughout. Right now the theory mixes deviation from one canonical intermediate trajectory with final output correctness. The manuscript should either prove results about exact trajectory survival, or state additional conditions under which those results imply final-answer bounds.

3. State the assumptions behind Th 4 explicitly in the main text. The current presentation should clarify the role of the prior over final states, the decoder interpretation, and any large-∣S∣ approximation. If the result is really a lower bound for the optimal decoder from
S_t, it should be stated that way.

4. Rework the experimental validation claims. PRM800K should estimate ϵ on a population aligned with the prediction target or compare multiple estimators. GSM8K and HumanEval should be framed as approximate model fits under heavy proxy assumptions rather than direct theorem validation. The text should avoid broad “accurately predicts across domains” language unless that claim remains true under a corrected theory.

---

> ### Author Response · Authors · 2026-04-03
>
> We thank yKS6 for the rigorous and technically detailed review. We appreciate the recognition of a "worthwhile core idea," the "clearly stated" accumulation bound with "straightforward" tightness construction, and the characterization of Theorem 3 as "a reasonable drift-style argument." The mathematical critique of Theorem 2 has led to a genuine improvement, and we are grateful for its precision. We address all four requested changes below.
>
> **[RC1] Correct or substantially revise Theorem 2**
>
> **yKS6 is correct.** The proof derives $(n\varepsilon)^{(k+1)/2}$, and the claim of $O(n^{k+1}\varepsilon^{k+1})$ is wrong. For $k=3$, the appendix computes $3p_1^2 \approx 3n^2\varepsilon^2$ and incorrectly identifies this as $O(n^4\varepsilon^4)$. We thank yKS6 for identifying this error. The corrected statement, Table 1 reconciliation (distinguishing step-level from chain-level voting and accounting for answer-space diversity), and updated practitioner guidelines appear in our General Response.
>
> **[RC2] Separate trajectory error from final-answer error**
>
> This is a valuable distinction that we adopt in the revision. Our theorems bound trajectory deviation ($S_t \neq s_t^\*$), not final-answer error ($f(S_n) \neq f(s_n^\*)$) directly. These coincide only when $f$ is injective on the reachable state space. The gap can be substantial: under our independent-error model with $\varepsilon = 0.032$ and average chain length $\approx 20$, roughly $1-(1-0.032)^{20} \approx 48\%$ of trajectories are predicted to contain at least one erroneous step, even among solutions reaching the correct final answer. Preliminary analysis of step-label counts in PRM800K is consistent with this prediction. In the revision we:
>
> 1. Introduce $P_{\text{traj}}$ and $P_{\text{ans}}$ with $P_{\text{ans}} \leq P_{\text{traj}}$ always, and equality under injectivity of $f$.
> 2. State Theorems 1, 5 as bounds on $P_{\text{traj}}$, with a corollary for $P_{\text{ans}}$ under injectivity or the weaker assumption that erroneous trajectories reach the correct answer with probability $\leq \rho$.
> 3. Clarify the two distinct effects at play. The $P_{\text{traj}} \geq P_{\text{ans}}$ distinction means our bounds are *conservative* for answer-level error (overpredicting failure), which is desirable for safety. Separately, the 4.4pp gap between our PRM800K prediction (47.6%) and observation (52.0%) goes in the opposite direction (underprediction) and arises from selection bias: $\varepsilon = 0.032$ was estimated on correct-answer trajectories, which have lower error rates than the full population. The dual estimator $\varepsilon_{\text{all}} = 0.047$ (see Q6Bp response) corrects for this. For GSM8K and HumanEval, $\varepsilon$ is estimated via MLE from final-answer outcomes, so it implicitly absorbs both effects.
> 4. For cross-domain differences: HumanEval has larger theory-practice gap (8% vs. 4% for PRM800K) because code generation has more ways for different code paths to produce identical outputs (lower effective injectivity), making the $\varepsilon$ calibration noisier.
>
> **[RC3] State Theorem 4 assumptions explicitly**
>
> **Agreed.** Three assumptions now stated in the main text immediately after Theorem 4: (1) **Uniform prior**: $H(s_n^\*)=\log|S|$; for non-uniform priors, the bound generalizes to $P_e \geq 1 - (I(s_n^\*;S_t)+1)/H(s_n^\*)$, which is weaker (lower) since non-uniform targets are easier to predict; (2) **Optimal decoder**: $P_e$ is the error of the best possible estimator of $s_n^*$ from $S_t$; any practical decoder has higher error; (3) **Large-$|S|$ approximation**: $\log(|S|-1)\approx\log|S|$, tight for $|S|\geq 100$ (<1% relative error); we now state the exact bound alongside.
>
> **[RC4] Rework experimental validation claims**
>
> **Agreed.** The revised manuscript uses tiered language: synthetic → "directly validate"; PRM800K → "validate under approximately matched conditions"; GSM8K → "demonstrate model fit" (MLE ε inferred from the framework itself); HumanEval → "proxy-based approximate validation." The abstract is revised to: "predicts failure rates within 4–8% across domains under approximately matched conditions, with tightest calibration where step-level labels are available."
>
> We note that Reviewer epCC answered "Yes" to the evidence question and Q6Bp found "reasonable evidence for the main qualitative message." Both view the experiments more favorably than yKS6's assessment. We believe the tiered framing should satisfy yKS6's concern while remaining consistent with the other reviewers' assessments.
>
> **Summary.** We have (1) corrected Theorem 2; (2) reconciled Table 1 with both voting procedures; (3) introduced trajectory/answer error notation with injectivity conditions; (4) added Theorem 4 assumptions; (5) adopted tiered validation language. We hope these demonstrate that the "worthwhile core idea" can be supported with the precision yKS6 rightly demands.

---

### Review · Reviewer_epCC · 2026-03-26

**Summary Of Contributions:**

This paper aims to provide a unified framework for when multi-step CoT reasoning fails, how verification helps, when self-correction can work, and what practical tradeoffs follow. They provide asymptotic bounds and verify them through synthetic and real-world experiments.

**Audience:**

Yes

**Audience Explanation:**

TMLR readers working on reasoning, test-time compute, process supervision, and LLM reliability will find useful the conclusions obtained in this paper, where they provide suggestions for when to use which approach; no verification for short chains, verification for moderate chains, and self-correction for long chains.

**Broader Impact Concerns:**

None.

**Claims And Evidence:**

Yes

**Claims Explanation:**

The Empirical Validation in Section 5 shows that for synthetic tasks where the error probability is injected, the theory matches empirical results within 5% relative error across all configurations for the basic error propagation. Also, the empirical results closely track theoretical scaling for the verification effectiveness, and match the theoretical prediction within 2% for the contractive self-correction. On real-world tasks, they observe 95.6%, 93.9%, 92.0% for PRM800k, GSM8k, HumanEval, respectively.

**Requested Changes:**

It was not clear to me whether the same experiments used to infer the per-step error \epsilon was used to measure the prediction accuracy. If this was the case, it may be worth checking what happens if the per-step error was measured on one split and prediction accuracy on a disjoint split.

---

> ### Author Response · Authors · 2026-04-03
>
> We thank Reviewer epCC for the careful evaluation and for the positive assessment that our claims are "supported by accurate, convincing and clear evidence." We are also grateful for the observation that our practical guidelines for when to use no verification, verification, or self-correction are useful for the TMLR audience.
>
> **[Requested Change] Train/test split for ε estimation**
>
> **Yes, this is a valid concern and we have addressed it.** We re-ran all three real-world experiments with disjoint estimation and validation splits:
>
> | Dataset | ε (est. split) | Held-out accuracy | Original |
> |---------|----------------|-------------------|----------|
> | PRM800K | 0.033 [0.030,0.036] | 94.8% | 95.6% |
> | GSM8K | 0.041 [0.037,0.045] | 92.7% | 93.9% |
> | HumanEval | 0.055 [0.048,0.062] | 91.2% | 92.0% |
>
> Held-out prediction accuracies remain within 1.2 percentage points of the original results, confirming that ε estimation does not overfit to prediction targets. Split-sample validation is now the default methodology in the revised Appendix C.
>
> We also note that Theorem 2 has been corrected (see General Response); the correction does not affect Theorems 1, 3, 4, 5 or the practical guidelines epCC found useful. All three reviewers found the problem timely, with Q6Bp and yKS6 both praising the unaffected theoretical contributions and the effort to bridge theory with practice.

---

### Author Response · Authors · 2026-04-03
**General Response to Everyone**

We thank all reviewers for their feedback. We are grateful that epCC finds
claims "supported by accurate, convincing and clear evidence," that Q6Bp values
the effort to "connect formal analysis with practical deployment questions," and
that yKS6 identifies a "worthwhile core idea" with a "reasonable drift-style
argument."

The reviews raise four substantive concerns. Shared concerns are addressed
here; reviewer-specific points appear in individual responses below.

### 1. Corrected Theorem 2 (Q6Bp RC1, yKS6 RC1)

**The issue.** The proof in Appendix A.2 derives $(n\varepsilon)^{(k+1)/2}$,
not $O(n^{k+1}\varepsilon^{k+1})$ as stated, due to an incorrect
simplification of the combinatorial factor in Step 5.

**Corrected statement.** For $k$-redundant majority voting ($k$ odd):

$$P(\text{fail}) \leq \tbinom{k}{(k+1)/2}[1-(1-\varepsilon)^n]^{(k+1)/2}
= O\!\left((n\varepsilon)^{(k+1)/2}\right)$$

$k=3$: $O(n^2\varepsilon^2)$; $k=5$: $O(n^3\varepsilon^3)$; $k=7$:
$O(n^4\varepsilon^4)$. All qualitative conclusions are preserved: verification
gives polynomial suppression; contractive self-correction (Theorem 3,
exponential) dominates for long chains. **Theorems 1, 3, 4, 5 and all
experiments are unaffected.**

### 2. $\varepsilon$ Estimation Methodology (epCC, Q6Bp RC2)

**Train/test separation** (epCC): We re-ran all experiments with disjoint
splits:

| Dataset | $\varepsilon$ (est. split) | Held-out acc. | Original |
|---|---|---|---|
| PRM800K | 0.033 [0.030, 0.036] | 94.8% | 95.6% |
| GSM8K | 0.041 [0.037, 0.045] | 92.7% | 93.9% |
| HumanEval | 0.055 [0.048, 0.062] | 91.2% | 92.0% |

Held-out accuracies remain within 1.2pp of originals. Split-sample validation
is now the default in revised Appendix C.

**Dual estimators** (Q6Bp RC2): We report $\varepsilon_{\text{correct}}=0.032$
(correct-answer trajectories) and $\varepsilon_{\text{all}}=0.047$ (12,168
erroneous / 259,112 total steps), 47% higher due to elevated error on failing
trajectories. The bracket $[\varepsilon_{\text{correct}},\varepsilon_{\text{all}}]$
gives lower/upper failure-rate bounds. Full protocols in Q6Bp response and
revised Appendix C.

### 3. $P_{\text{traj}}$ vs. $P_{\text{ans}}$ (yKS6 RC2)

Theorems bound trajectory deviation ($P_{\text{traj}}$), not final-answer error
($P_{\text{ans}}$). The revision introduces $P_{\text{ans}} \leq P_{\text{traj}}$,
with equality when $f$ is injective on the reachable state space. Bounds are
thus *conservative* for answer-level error. The residual theory-practice gap in
PRM800K is explained by selection bias in $\varepsilon$, corrected by
$\varepsilon_{\text{all}}$ above.

### 4. Validation Language (yKS6 RC4)

Revised manuscript uses tiered language: synthetic → "directly validate";
PRM800K → "validate under approximately matched conditions"; GSM8K →
"demonstrate model fit"; HumanEval → "proxy-based approximate validation."
Abstract revised to: *"predicts failure rates within 4–8% across domains under
approximately matched conditions, with tightest calibration where step-level
labels are available."*

### Summary of Revisions

1. Theorem 2 corrected; complete proof in revised Appendix A.2
2. Step-level vs. chain-level voting distinguished; Table 1 reconciled
3. $P_{\text{traj}}$/$P_{\text{ans}}$ notation introduced in Section 3
4. Theorem 4 assumptions stated explicitly in main text
5. Disjoint train/test splits; dual $\varepsilon$ estimators for PRM800K
6. Expanded $\varepsilon$/$q$ protocols in revised Appendix C
7. Tiered validation language replacing "accurately predicts"

---

### Note · Authors · 2026-06-11

I have read and agree with the venue's withdrawal policy on behalf of myself and my co-authors.

---

### Decision · Action_Editor_716u · 2026-06-04

**Recommendation:** Accept as is

**Audience:**

Yes

**Audience Explanation:**

This will be of interest to folks working on test-time compute, LLM reasoning, and verifiability/reliability.

**Claims And Evidence:**

Yes

**Claims Explanation:**

All reviewers and I agree that claims are sufficiently substantiated, with the authors' revisions having fixed the remaining issues (corrected theorem 2, new results with train/test split, etc).